# The Added Value of Point-Light Display Observation in Total Knee Arthroplasty Rehabilitation Program: A Prospective Randomized Controlled Pilot Study

**DOI:** 10.3390/medicina58070868

**Published:** 2022-06-29

**Authors:** Christel Bidet-Ildei, Quentin Deborde, Victor Francisco, Elise Gand, Yannick Blandin, Anne Delaubier, Anne Jossart, Philippe Rigoard, Maxime Billot, Romain David

**Affiliations:** 1Université de Poitiers, Centre de Recherches sur la Cognition et l’Apprentissage, 86000 Poitiers, France; victor.francisco@univ-poitiers.fr (V.F.); yannick.blandin@univ-poitiers.fr (Y.B.); 2Service de Médecine Physique et Réadaptation, Centre Hospitalier Universitaire de Poitiers, 86000 Poitiers, France; quentin.deborde@gmail.com (Q.D.); anne.delaubier@chu-poitiers.fr (A.D.); anne.jossart@chu-poitiers.fr (A.J.); romain.david@chu-poitiers.fr (R.D.); 3CHU Poitiers Clinical Investigation Center CIC 1402, INSERM, University of Poitiers, 86000 Poitiers, France; elise.gand@chu-poitiers.fr; 4PRISMATICS (Predictive Research in Spine/Neurostimulation Management and Thoracic Innovation in Cardiac Surgery), Poitiers University Hospital, 86000 Poitiers, France; philippe.rigoard@univ-poitiers.fr (P.R.); maxime.billot@chu-poitiers.fr (M.B.); 5Department of Spine Surgery and Neuromodulation, Poitiers University Hospital, 86021 Poitiers, France; 6Pprime Institute UPR 3346, CNRS, ISAE-ENSMA, University of Poitiers, 86360 Chasseneuil-du-Poitou, France

**Keywords:** point-light display, action observation, rehabilitation, total knee arthroplasty

## Abstract

*Background and Objectives*: The present study aimed to assess the potential benefit of the observation of rehabilitation-related point-light display in addition to a conventional 3-week rehabilitation program, the objective being to improve functional capacity in patients having undergone total knee arthroplasty. *Materials and Methods*: Patients randomized in the control group had conventional rehabilitation treatment with two sessions per day 5 days a week of physical therapy (90 min), whereas patients in the experimental group had a program of conventional rehabilitation combined with a point-light display observation two times per day (5 min) and 3 days a week. *Results*: The patients of both groups had improved their performances by the end of the program, and the pre- and post-test improvement were superior for the experimental group over the control group concerning the total WOMAC score (*p* = 0.04), the functional WOMAC score (*p* = 0.03), and correct recognition of point-light displays (*p* = 0.003). *Conclusions*: These findings provide new insight favoring systematic point-light display observation to improve functional recovery in patients with total knee arthroplasty.

## 1. Introduction

Total knee arthroplasty (TKA) nowadays represents an international standard of care with 1,324,000 total knee primary and revision procedures in 18 countries worldwide, especially in the aging population [1]. Despite major technological and technical advances to optimize TKA surgery [2], a key challenge is still the rehabilitation from impaired mobility, which has been shown to hamper daily life activity, social participation, and quality of life [3,4]. While conventional rehabilitation provided by physiotherapists is widely used, some new solutions/approaches should be explored in order to potentiate the walking capacity recovery in persons having undergone TKA, such as the observation of videos reproducing specific knee movements.

By activating mirror neuron networks [5,6], Action Observation Training (AOT) has been used for more than a decade to improve motor function rehabilitation in pathologies with central deficits such as stroke [7,8], cerebral palsy [9,10], and Parkinson’s disease [11]. While AOT efficacy has been clearly proven for rehabilitation of central disorders [12], this approach has been significantly less studied in peripheral disorders [13,14,15]. In a randomized case-control study, Bellelli et al. [13] used a 3-week combined program including conventional post-orthopedic rehabilitation (1 h a day, 6 days a week) and AOT (24 min a day, 6 days a week) in 60 patients with post-orthopedic surgery (hip arthroplasty, knee arthroplasty, hip fracture repair). The authors reported benefits of adding AOT combined with a conventional rehabilitation program to improve functional capacity (assessed with Functional Independence Measure (FIM)). Similarly, improvement of functional capacity, assessed with the Western Ontario and McMaster Universities Osteoarthritis Index (WOMAC), has been observed in 18 TKA patients participating in a 3-week rehabilitation program combining conventional rehabilitation (30 min a day, 3 days a week) and AOT (10 min a day, 3 days a week) [14]. Therefore, AOT may be able to limit the alteration of movement and motor performance induced by limb non-use [16]. Moreover, we hypothesize that the reactivation of healthy and painless motor pattern with action observation could facilitate motor capacity recovery through a top-down effect [13]. All in all, these studies provide encouraging results, suggesting AOT as a complementary approach to potentiate conventional rehabilitation programs [12].

In addition to the beneficial aspects, AOT is simple to implement insofar as it requires little equipment, few staff members, and can be performed at different times outside of care. However, AOT requires a massive videos database or real-time observation of a participant, which is relatively costly. Moreover, each author classically builds his/her stimuli to assess the effects of AOT for specific rehabilitation programs, which renders it difficult to standardize in clinical practice. One alternative could be the use of point-light displays (PLD).

With limited information presented to a participant, numerous studies have demonstrated that with the PLD technique [17], which consists of displaying a basic reconstructed model with a joint visible to observers, he/she is able to identify a movement [18,19,20,21,22,23]. PLD has been largely used in the literature (i) to investigate the mechanisms involved in action observation (e.g., [24,25,26]) and (ii) to learn new motor gestures [27,28,29,30,31]. For example, Saber et al. [31] have compared the efficacy of classical action observation and PLD observation in the acquisition of new motor skills in children with Autism Spectrum Disorder. They showed that both classical observation and PLD observation are effective ways of learning new motor skills. Interestingly, the condition with PLD observation has entailed more attention to the relevant limbs than the condition with classical observation. In the same manner, Francisco et al. [28] have shown that PLD observation can improve biomechanical and transfer performance in judo practice. Consequently, by adding a simplified approach to AOT, PLD could reinforce previous promising results observed in the walking capacity of TKA patients after a combined rehabilitation program.

In order to obtain new information on the walking capacity management of TKA with AOT, we designed a prospective randomized pilot study comparing conventional therapy over a 3-week period with or without observation of PLD for the recovery of motor function in patients with TKA. The primary objective was to assess whether the addition of PLD observation to conventional rehabilitation can improve the functional mobility (Time Up and Go test) of patients with total knee arthroplasty. The secondary objectives were to compare evaluation of PLD observation on motor recovery (pain, stiffness, and the difficulty of daily life actions) and action recognition. Our main hypothesis was that the addition of PLD observation should improve the functional recovery of patients with total knee arthroplasty, as has been observed for classical action observation [13,14].

## 2. Materials and Methods

### 2.1. Study Design

This two-arm parallel prospective interventional randomized controlled monocentric pilot study was designed to assess the added value of PLD training combined with conventional rehabilitation of walking in patients presenting with TKA after 3 weeks.

Recruitment was performed in the Physical Medicine and Rehabilitation department of the University Hospital of Poitiers between 2019 and 2021. All participants gave their informed written consent before their inclusion in the study. The study was conducted in accordance with the Declaration of Helsinki and approved by the ethics committee. The study was approved by the CPP Ile de France II (2019-A00450-57) and is registered at clinicaltrials.gov (NCT03856983).

### 2.2. Inclusion and Exclusion Criteria

All volunteers over 18 years of age having undergone knee surgery could be included in the study. Patients could not be included in the study if they had any locomotor condition not due to knee surgery, uncorrected visual disturbances, or comorbidity altering locomotion (history of stroke, neurological condition, inflammatory rheumatism).

In addition, participants were allowed to continue if MMSE > 21 [32] and BDI < 9 [33]. These non-inclusion criteria are similar to those applied in the studies by Villafane and Belleli [13,15].

After written consent, randomization was made and a group (experimental or control) was attributed to each patient. Following which, each patient underwent initial evaluation (motor and perceptual tasks) and three weeks of rehabilitation in accordance with his/her group. At the end of rehabilitation, all patients underwent the final evaluations (motor and perceptual tasks).

### 2.3. Procedure

Eligible patients were randomly assigned in a 1:1 ratio to the control or the PLD group. Randomization was automatically made with a stratification by age (18–28 years, 29–60 years, and ≥61 years) and sex (male/female) in accordance with the influence of these parameters on visual perception of PLDs [21,34].

The control group received conventional rehabilitation treatment with 2 sessions of 90 min of physical therapy per day, 5 days a week for 3 weeks. Conventional rehabilitation consisted of physiotherapy sessions with passive and active (with an arthromotor) mobilization of the knee focusing on range of motion (techniques for awakening the quadriceps, active work assisted by flexion of the knee), massages, contract-release techniques, and balneotherapy with healing waters.

The PLD group received similar conventional rehabilitation combined with a 5 min PLD observation and recognition tasks before and after physical therapy for 1 session per day (on the morning) for 3 days a week. PLD observation and recognition tasks consisted of observing an animated sequence representing human movement of the lower limbs for 20 s (see Figure 1) and orally naming the presented action within 3 s. Since many works have shown that the simple observation of the PLD was sufficient to activate the mirror neuron system [35,36], the recognition task performed was considered as a control condition to ensure that patients were focused on the PLD.

Visual stimuli consisted of PLD representing human movements displayed centered in a tablet screen (10′, 25 cm in diagonal). These PLDs are built from the recording of a young man who performed the different actions with 33 markers on his body (see Appendix B for a complete description). Each action was captured with a Vicon-Nexus system at 100 htz of sample rate with 20 MX-T40 cameras. Afterwards, each stimulus was modified withPLAViMoP softwares 1 and 2, creator: Beauprez, S-A., Bidet-Ildei, C., Blandin, Y., Decatoire, A., Lacouture, P., & Pylouster, J. Poitiers, France APP N° 1 (2016) IDDN.FR.001.200011.000.S.P.2017.000.31235. APP N°2: APP N° (2021) IDDN.FR.001.200011.001.S.A.2017.000.31235. [37]. Following transformations, all stimuli were composed of 13 dots representing the main limbs of the body (shoulders, elbows, wrists, hips, knees, ankles) and the head. The size of the dots was 0.5 cm and the zoom applied produced a PLD about 15–20 cm high on the screen. For the perceptual tasks, twenty-four PLDs (12 representing lower limb actions and 12 representing upper-limb action, see Appendix A for a table with all actions carried out) were collected on the PLAViMoP platform (https://plavimop.prd.fr/en/motions, accessed on 1 July 2019). For the observation task, the 12 lower-limb PLD human actions were used. For both perceptual and observation tasks, all PLDs were presented centered in the screen. Once again, the patient had to name the action within 3 s. If the patient did not know the response or provided a wrong answer (e.g., jump instead of walk), the experimenter gave the correct answer after 3 s.

The procedure was divided into 3 phases: initial evaluation, rehabilitation, and final evaluation (see Figure 2).

### 2.4. Primary Outcome

The “Time Up and Go” test (TUG, [38]) assessed the patients’ functional mobility. This test evaluates the patient’s ability to rise from a chair, walk 3 m, come back to the chair, and sit down. Time was measured before and after the 3-week rehabilitation program.

### 2.5. Secondary Outcomes

Motor outcome was assessed with the WOMAC questionnaire (Western Ontario and McMaster Universities Osteoarthritis Index, [39]). Frequently used for follow-up of patients who have undergone surgery [14], it assesses pain, stiffness, and the difficulty of daily life actions. These three measures were assessed with a 5-point Likert scale (0 representing a zero level and 4 representing an extreme level, see Appendix C) before and after the 3-week rehabilitation program.

Perceptual tests completed before and after the 3-week rehabilitation program consisted of a PLD recognition task (see Figure 3) comprising 24 animated sequences representing human actions (12 lower limb actions and 12 upper limb actions) over 3 s. After each presentation, the patient was asked to orally name the presented action within 3 s. Patients scored 1 in case of right response or 0 in case of wrong or no response. No feedback on the correctness of the answers was provided.

### 2.6. Statistical Analysis

Variables were checked for the respect of normality (with the Shapiro–Wilk test) and variance homogeneity (with the Levene test). Times to perform the TUG before and after the rehabilitation program did not respect normality. Consequently, comparison of the experimental and the control groups was performed with a non-parametric Mann–Whitney test. For the other tasks (WOMAC and recognition scores), a mixed ANOVA was used where the group (control, PLD) was considered as grouping factor and the time (pre, post) was considered as repeated factor. When interaction was significant, post hoc comparisons were performed with Bonferroni tests. Moreover, to specifically compare the improvement for control and experimental groups, we completed our analysis with an ANCOVA by using the difference between the performance in post-test and the performance in pre-test as dependent variable, the group (control, PLD) as fixed factor, and the performance in pre-test as covariate. For this last analysis, we used unilateral comparisons with the hypothesis that the experimental group should have better improvement than the control group. For all analyses, the significance level was set at *p* < 0.05. We considered that the effect was tendential when p was inferior to 0.10. For each significant analysis, partial eta square indicates effect size. 

## 3. Results

### 3.1. Participants

Thirty-six adults (18 females, 62.9 ± 8.7 years) who underwent a total knee arthroplasty were included. Three participants were excluded during the study (one for COVID-19, one for infection, and one for fall with fracture). Moreover, the data of five participants were removed from the analysis because their scores in pre- and/or post-test were too far from the mean of their group (more than 1.6 standard deviation). No participant presented cognitive disorders, depression, uncorrected visual disturbances, or comorbidity altering locomotion (history of stroke, neurological condition, inflammatory rheumatism). Finally, 14 participants remained in the control group and 14 participants in the experimental group (Table 1).

### 3.2. Functional Mobility

The difference between pre- and post-test times in TUG were compared for the two groups with a Wilcoxon test. The results indicate an improvement between pre- and post-test performances in TUG in the control (M = −11.5 ms, SD = 15.2 ms, Kendall W = 0.85; *p* < 0.001) and in the experimental group (M = −15.3 ms, SD = 14.2 ms, Kendall W = 0.74; *p* < 0.001).

Mann–Whitney comparison showed that this improvement was not different between groups (U14,14 = 124; *p* = 0.25, also see Figure 4 and Figure 5A).

### 3.3. Motor Outcome

Analysis of the global WOMAC score (see Figure 6A) showed no significant main effect of group (F (1,26) = 1.21; *p* = 0.28), while a significant main effect of time (F (1,26) = 55.04; *p* < 0.001; η^2^P = 0.68) and interaction between the group and time (F (1,26) = 4.67; *p* < 0.05; η^2^P = 0.15) was observed. Post hoc analysis indicated that patients in both groups improved their scores (*p* < 0.01 in both cases). The ANCOVA showed an effect on the pre-test (F (1,25) = 33.31; *p* < 0.001; η^2^P = 0.57) and a tendential effect of the condition (F (1,25) = 2.26; *p* = 0.08), which is characterized by greater improvement for the experimental group than for the control group (see Figure 7A). 

Analysis of the WOMAC pain score (see Figure 6B) showed no main effect of group (F (1,26) = 1.64; *p* = 0.21), a main effect of time (F (1,26) = 15.67; *p* < 0.001; η^2^P = 0.38), and no significant interaction between group and time (F (1,26) = 1.64; *p* = 0.21). The time main effect indicates that both groups reduced their pain score from pre-test (M = 8.96; SD = 4.68) to post-test (M = 6.64; SD = 3.12). The ANCOVA showed an effect on the pre-test (F (1,25) = 28.92; *p* < 0.001; η^2^P = 0.54) but no effect of the condition (F (1,25) = 0.10; *p* = 0.38), which suggests that the improvement between post-test and pre-test is similar in both groups (see Figure 7B).

Analysis of the WOMAC stiffness score (see Figure 6C) showed no main effect of the group (F (1,26) = 0.35; *p* = 0.56), a main effect of time (F (1,26) = 17.27; *p* < 0.001; η^2^P = 0.40), and no interaction between group and time (F (1,26) = 0.23; *p* = 0.63). Patients in both groups reduced their stiffness score from pre-test (M = 4.54; SD = 1.92) in comparison with post-test (M = 3.32; SD = 1.59). The ANCOVA showed an effect on the pre-test (F (1,25) = 13.28; *p* < 0.001; η^2^P = 0.34) but no effect of the condition (F (1,25) = 0.01; *p* = 0.46), which suggests that the improvement between post-test and pre-test is similar in both groups (see Figure 7C).

Analysis of the functional WOMAC score (see Figure 6D) showed no main effect of group (F (1.26) = 1.04; *p* = 0.31), a main effect of time (F (1,26) = 54.13; *p* < 0.001; η^2^P = 0.68), and a significant interaction between group and time (F (1,26) = 5.12; *p* < 0.05; η^2^P = 0.16). Post hoc analysis showed that both groups improved their functional score from pre-test (M = 20.21; SD = 10.6) to h post-test (M = 11.86; SD = 7.25). The ANCOVA showed an effect on the pre-test (F (1,25) = 26.31; *p* < 0.001; η^2^P = 0.49) and a tendential effect of the condition (F (1,25) = 2.59; *p* = 0.06), which is characterized by greater improvement for the experimental group than for the control group (see Figure 7D).

### 3.4. Perceptual Evaluation

The analysis of the scores of correct PLD recognition (Figure 4B) showed no main effect of group (F (1,26) = 0.01; *p* = 0.91), a main effect of time (F (1,26) = 37.52; *p* < 0.001; η^2^P = 0.59), and a significant interaction between group and time (F (1,26) = 13.35; *p* < 0.001; η^2^P = 0.37). Post hoc analysis indicated that only patients in the experimental group benefited (*p* < 0.001), whereas patients in the control group did not (*p* = 0.78). The ANCOVA showed an effect on the pre-test (F (1,25) = 12.65; *p* = 0.002; η^2^P = 0.34) and an effect of the condition (F (1,25) = 13.80; *p* =0.001; η^2^P = 0.36), which is characterized by greater improvement for the experimental group than for the control group (see Figure 5B). 

## 4. Discussion

The objective of this study was to determine whether PLD observation added to a conventional rehabilitation program could improve the recovery of patients with total knee arthroplasty. The results showed that the addition of the PLD observation had no effect on TUG performance, pain, and stiffness of the WOMAC, whereas significantly or tendential greater benefit was observed in the experimental group compared to the control group on functional scores, global WOMAC scores, and action recognition.

As was reported by Park et al. [14], the benefit obtained from action observation was shown for WOMAC evaluation but not for TUG performance. However, our results differ on pain and stiffness outcomes, where we did not obtain an interaction between group and time of evaluation. For pain, this could be explained by our not using the same evaluation (VAS for [14], and WOMAC in the present study). For stiffness, the difference could be due to the patients who were included in the studies. Indeed, in the present study, the patients had a lower level of stiffness score before the rehabilitation program (mean = 4.5) than in the study by Park et al. [14] (mean = 7.1). Similarly, the absence of significant difference between experimental and control groups in the post-test (all *p* > 0.08) could be due to the pre-test scores. For instance, the patients in the current study had an average score of 20.2, whereas the patients in the Park study had an average score of 73. Moreover, even if the PLD observation practice was similar to the Park study, the physical practice time was three-fold greater in our study (2 × 90 min/day, 5 days a week vs. 30 min/day, 3 days a week), which could have increased the benefits obtained by the control group. Nevertheless, in the present study, the tendential differences between evolution of the control and the experimental groups for the global WOMAC scores and the functional WOMAC scores suggest greater improvement in the experimental than in the control group. This implies that PLD observation could induce positive effects on functional recovery such as results with classical AOT [13,14], especially in patients having undergone total knee surgery. Moreover, our study showed that PLD observation improves the capacity to recognize PLD actions, suggesting that sensitivity to PLD can improve with training. Consequently, PLD observation can be considered as a valuable addition to classical AOT in the functional recovery of patients suffering from locomotor disorders.

In accordance with the literature, we can hypothesize that the mechanism involved in PLD observation results from a “top-down effect” similar to classical AOT [13]. Previous studies have shown that PLD observation activates motor representations at the central level [36,40,41,42], which could in turn lead to peripheral modification. Therefore, we can suggest that PLD observation was able to activate the motor system via the mirror neuron mechanism and consequently limit the alteration of movement and motor performance caused by limb non-use while preserving the cortical organization of motor system [16]. One alternative or additional hypothesis could be that PLD observation activates the healthy and painless motor patterns of patients (i.e., prior to the surgery and the onset of pain), which could facilitate motor recovery. These hypotheses should be tested in future neuroimaging studies.

The main limitation of our study is the heterogeneity of our group in the pre-test. Despite the randomization of our patients in the control and the experimental groups, there is a difference in the pre-test for all our variables that increases the variability and could explain why some of our effects are only tendential and not significant. The second limitation is the absence of a visual control condition that would have had video observation on a tablet such as e-landscapes [43] or video clips without motor content [13]. Moreover, if the WOMAC index is a reference evaluation for patients with knee arthroplasty, this subjective assessment needs to be combined with objective tools. Finally, future studies should be made to confirm these results and to more precisely determine how PLD observation of PLD can improve the rehabilitation of patients with locomotor disorders. In addition, focusing specifically on the main joint involved in an action (e.g., legs for a jump) and/or the sex of patients related to the sex of the AOT could help to provide a personalized AOT rehabilitation program.

## 5. Conclusions

In conclusion, the present study showed that when added to a conventional rehabilitation program, PLD observation can improve the functional recovery of patients with locomotor disorders. These findings provide new perspectives to better define the effect of action observation in motor rehabilitation and to consider its use in daily practice activities.

## Figures and Tables

**Figure 1 medicina-58-00868-f001:**
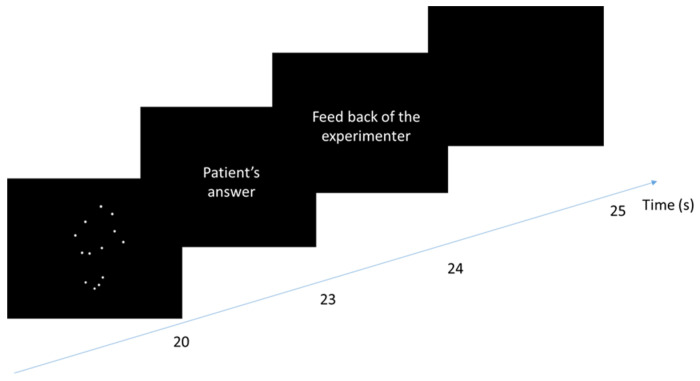
Time course of a trial of observation task for the experimental group. The patients saw on a loop (over 20 s) a PLD representing a lower human action. After that, they had 3 s to name the perceived action. When the response was correct, the experimenter said, “it is correct”, and when it was false, the experimenter gave the correct answer. The next trial started after 1 s.

**Figure 2 medicina-58-00868-f002:**
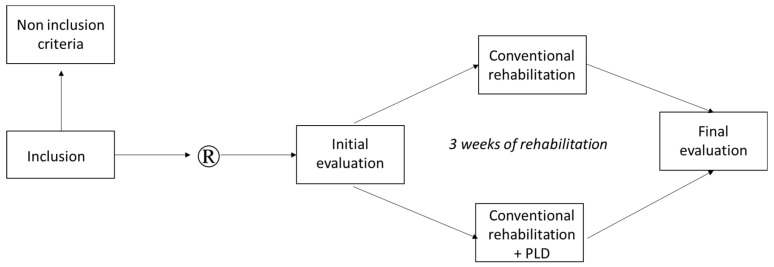
Experimental design.

**Figure 3 medicina-58-00868-f003:**
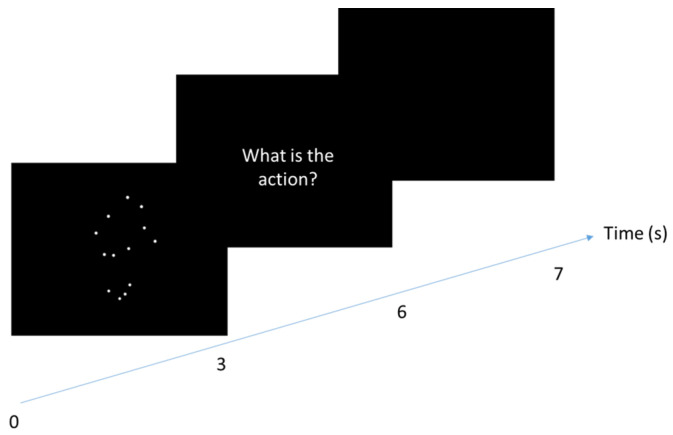
Time course of a trial of the recognition task carried out by both groups in initial and final evaluations. The patients saw a PLD representing an upper or a lower human action (from 1 to 3 s) and they had to name the action perceived. They had 2 s to answer. For each trial, the experimenter noted whether or not the answer was correct.

**Figure 4 medicina-58-00868-f004:**
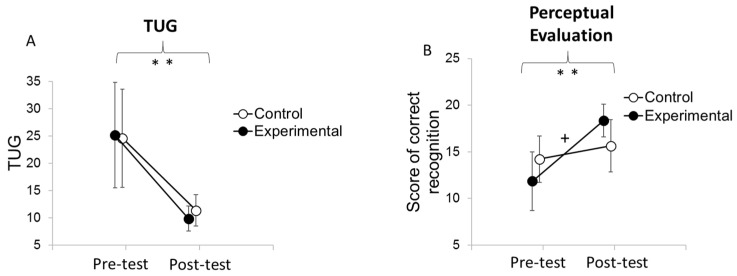
Mean results of TUG and perceptual evaluations by group before and after the rehabilitation program. Error bars represent confidence interval at 95%. ** indicates a difference between pre and post-test at *p* < 0.001. “+” indicates an interaction between the moment of the evaluation (pre-test, post-test) (**A**) and the group (control, experimental) (**B**) at *p* < 0.05. Pre-test indicates the level of performance at 0 weeks (before the rehabilitation program), and post-test indicates the level of performance at 3 weeks (at the end of the rehabilitation program).

**Figure 5 medicina-58-00868-f005:**
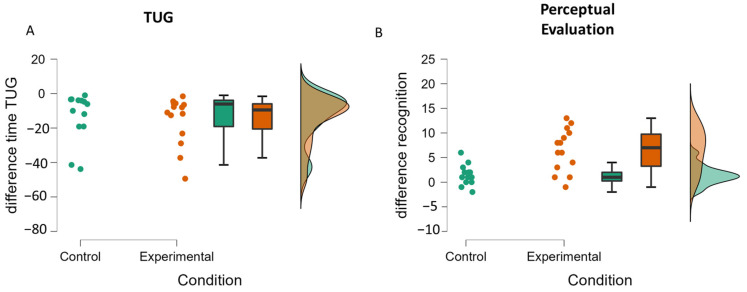
Raincloud plots of the difference between pre-test and post-test for the TUG (**A**) and the perceptual evaluation (**B**) for the control and the experimental groups.

**Figure 6 medicina-58-00868-f006:**
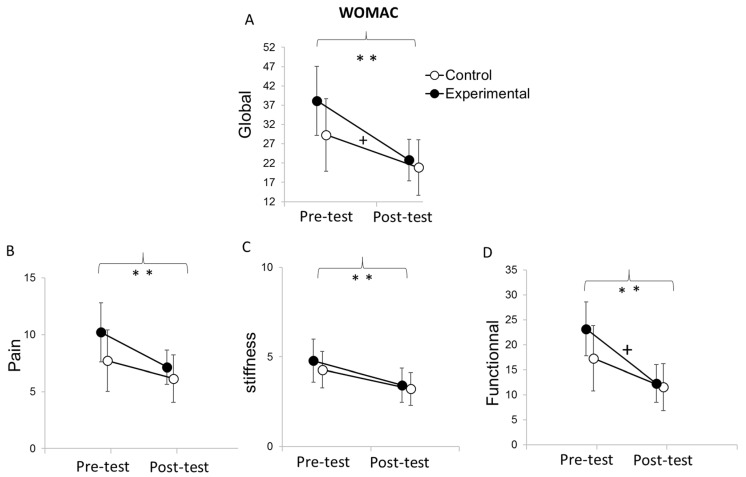
Mean results of WOMAC scores for pre-test and post-test and for the control and the experimental groups (**A**–**D**). Error bars represent confidence interval at 95%. ** indicates an effect at the moment of the evaluation at *p* < 0.001. + indicates an interaction between the moment of the evaluation and the condition at *p* < 0.05. Pre-test indicates the level of performance at 0 weeks (before the rehabilitation program), and post-test indicates the level of performance at 3 weeks (at the end of the rehabilitation program).

**Figure 7 medicina-58-00868-f007:**
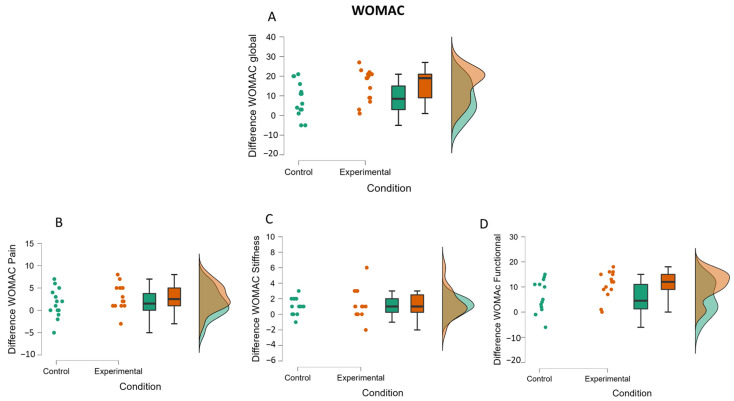
Raincloud plots of the difference (between pre-test performance and post-test performance) in WOMAC (global, pain, stiffness and functional) (**A**–**D**) for the control and the experimental groups.

**Table 1 medicina-58-00868-t001:** Characteristics of patients included in the control and experimental groups and inter group comparisons assessed with independent Student *t*–test.

Variables	Control Group	PLD Group	*p*-Value
Age (years)	60.4 ± 10.3	64.8 ± 6.4	0.192
BMI (kg/m²)	30.7 ± 4.8	29.9 ± 4.8	0.691
Score VAS	2.8 ± 2.2	3.9 ± 1.9	0.149
MMSE	27.6 ± 2.4	28.8 ± 1.3	0.127
Beck depression score	1.07 ± 1.73	1.00 ± 2.45	0.930

* means that the PLD was used in the task.

## Data Availability

The data presented in this study are available on request from the corresponding author. The data are not publicly available due to the French regulation for the protection of health data.

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
