# Peer review of "The Added Value of Point-Light Display Observation in Total Knee Arthroplasty Rehabilitation Program: A Prospective Randomized Controlled Pilot Study"

_medicina, 2022, doi:10.3390/medicina58070868_

Round 1
Reviewer 1 Report
The authors postulated that the effect of AOC could be transferred to TKA. Since displaying motions via PLD is more convenient and cost less than the videos, the authors investigate the effect of observing PLD motion in TKA rehab. They measured the time up to go test, WOMAC index, and accuracy of PLD recognition and then compared the performance between control and experimental groups. The experimental group performed better than the control group in the WOMAC and PLD recognition test in the post-test.
1. The authors need to provide the reason why the AOC can also work for TKA. AOC can facilitate neuroplasticity adaptation due to the activation of mirror neuron cortical networks. Since TKA is not a neuro disease, what are the mechanisms that the AOC might be useful for training TKA patients? I don’t see causation between AOC and TKA and hard to be persuaded the improvement observed in the experiment group is not by chance or other controlled factors.
2. AOC requires the participant to observe a motor task and then perform the same task. Why does the experimental design only require the patent to identify different movements? Identifying movement is related to cognitive training which is not part of the comorbidity or complication for TKA. Please explain the logic of the experimental design.
3. It is not clear what benefit the TKA patients can get from the PLD intervention and how the functional improvement can be observed from the evaluation. The authors should explain the selection of the measurements and how these tests related to the PLD intervention.
4. From figure 5, the experimental group showed a significantly worst performance in the pre-test. It indicates the failure of the randomized control and makes it hard to claim the effect of PLD intervention. Since the experiment has been done, I suggest the authors at least re-run the statistic analysis by controlling the initial performance.
5. In lines 327-330, the authors hypothesize the mechanism in PLD. “PLD observation activates motor representations at the central level [e.g., 36–39] which could, in turn, lead to peripheral modification.” A deeper discussion related to the finding of WOMAC scores can improve the contribution of the paper. How the peripheral modification could potentially reduce pain, stiffness, etc.
Author Response
Dear reviewer,
thank you for your comments. You find in the attachment a point-by-point response.
Sincerely yours

Reviewer 2 Report
The study is very interesting and novel.
As aspects to improve, better explain figures 2 and 3, since they provide little information, and make them smaller.
I suggest adding this current reference in the introduction:doi: 10.3390/medicina58020227
On the other hand, regarding the analysis of results, it is necessary to carry out Ancova in a longitudinal study, for all the variables. Version 28 of spss allows non-parametric Ancova. Once the differential score of the post, minus the pretest, is established, use this differential score as a dependent variable, and the pretest score as a covariate, to control that there are no previous differences.
Once these analyzes are done, rewrite results, and I will reconsider.
Thank you.
Author Response
Dear reviewer,
Thank you for your comments. You find in the attachment a point-by-point response file.
Sincerely yours

Reviewer 3 Report
From my point of view, the work is well-done and provides new insight favoring systematic point-light display observation to improve functional recovery in patients with total knee arthroplasty and thus . Just, I suggest some minor language modifications .
Author Response
Dear reviewer,
Thank you for your comments. You find in the attachment a point-by-point response file.
sincerely yours

Round 2
Reviewer 2 Report
The name of tables 2, 3 and 4 cannot be that long. add that text in results if necessary.
In table 3, I do not understand the value of the difference, how do you get
I still think the same, an Ancova is the appropriate analysis. The t test has no control over the pretest.
The statistics of the article do not reflect the results.
It is an interesting article, with a wrong statistic.
Author Response
Thank you for your comment. You will find our responses in the attached files. We sincerely hope that our new approach is now in line with your expectations.
